# “Youth Are More Aware and Intelligent than Imagined”: The Mountain Air Youth Photovoice Project

**DOI:** 10.3390/ijerph16203829

**Published:** 2019-10-11

**Authors:** Kathryn M. Cardarelli, Marcy Paul, Beverly May, Madeline Dunfee, Steven Browning, Nancy Schoenberg

**Affiliations:** 1College of Public Health, University of Kentucky, Lexington, KY 40506, USA; beverly.may@uky.edu (B.M.); madeline.dunfee@uky.edu (M.D.);; 2Center for Health Equity Transformation, University of Kentucky, Lexington, KY 40506, USA; Nesch@uky.edu; 3School of Public Health, University of North Texas Health Science Center, Fort Worth, TX 76107, USA; marcy.paul@unthsc.edu; 4College of Medicine, University of Kentucky, Lexington, KY 40506, USA

**Keywords:** Appalachia, youth, respiratory, community-based participatory research (CBPR), photovoice, environmental health, health disparities

## Abstract

Appalachian Kentucky reports some of the highest rates of respiratory illness in the United States, including chronic obstructive pulmonary disease and asthma. While smoking rates are high in the region, unexplained variation remains, and community-engaged research approaches are warranted to identify contributing factors. The Mountain Air Project’s community advisory board recommended that investigators invite youth to provide their perspectives on possible contributing factors to respiratory illness, and we undertook an exploratory study to determine the utility of photovoice to elicit such perspectives with this population. While photovoice has been employed for other youth-focused health studies in Appalachia, to our knowledge, this work represents the region’s first environmental study using photovoice among youth. Over eight weeks, ten participants (age 12–18) represented their perspectives through photographs and accompanying narratives. A brief thematic content analysis of the youth narratives that accompanied the photos revealed three primary themes of environmental determinants of respiratory illness. These themes included compromises community members make regarding respiratory health in order to secure a livelihood; tension between cultural legacies and respiratory health; and consequences of geographic forces. This study demonstrates the value of incorporating youth perspectives in environmental health research, and that photovoice was a valuable approach to elicit such perspectives.

## 1. Introduction

Respiratory illness rates in southeastern Kentucky are among the highest in the United States (US). According to the Centers for Disease Control and Prevention (CDC), the national asthma prevalence among adults in the US is 7.7%, compared to 10.7% in Kentucky [1]. Smoking is a significant contributing factor to respiratory morbidity and mortality. Kentucky has the second highest smoking prevalence among adults, 24.6%, compared to the US prevalence, 14.0% [2]. Although this higher prevalence of smoking in Kentucky contributes to the excess respiratory morbidity, a great deal of unexplained variation in respiratory disease remains.

To better understand contributors to this elevated respiratory disease rate, a team of investigators examined potential environmental determinants of respiratory illness in Appalachian Kentucky. The study, entitled “Mountain Air Project” (MAP) by community members, drew extensively on the engagement of a community advisory board (CAB) comprised of local residents. The primary goals of MAP were to conduct an epidemiologic study to examine the relationships among (1) environmental exposures, in particular indoor and outdoor pollutant sources, and (2) behavioral and social determinants and their cumulative effects on the risk of respiratory disease, especially asthma and chronic obstructive pulmonary disease (COPD), among adults in rural Appalachia. In addition, the MAP team aimed to distribute results of the study to a broad array of community stakeholders. We hoped to enhance understanding of the interaction of multiple factors on the risk of respiratory disease and to develop a targeted disease self-management and home modification intervention for the community. The final stage of the project (ongoing) involves the evaluation of the community-based intervention. Additional information is provided elsewhere [3]. During a CAB retreat in 2017, MAP CAB members suggested that investigators incorporate the perspectives and engagement of youth. Therefore, an exploratory study was implemented in 2018 to obtain grounded perspectives on what youth considered contributors to respiratory disease and to gauge their awareness of perspectives on well-known and emerging contributors using a community-based participatory research (CBPR) approach called photovoice.

Photovoice is a research approach that equips community residents with cameras and journals to capture their everyday experiences and explanatory narratives to generate knowledge about priority health issues [4]. Developed by Caroline Wang and colleagues, photovoice can provide investigators with the community lens on a variety of health topics and has been implemented in multiple populations [5,6]. This local contextualization of priority health concerns and perceived contributing risks and protective factors engages community members in the learning and understanding of complex health challenges and potential solutions.

Whereas photovoice has been used in previous studies with Appalachian youth [7,8,9], to our knowledge, this study is the first to use the approach to address environmental health. This paper provides findings from an exploratory study in which youth identified factors that they perceived as important environmental determinants of respiratory illness in their community.

## 2. Materials and Methods

The objective of this photovoice project was to engage and encourage youth to share their perspectives on environmental determinants of respiratory illness and involve them as catalysts for environmental health change in their community. With roots in critical Freirian education, feminist theory, and other photo-elicitation traditions, photovoice is a CBPR approach valued for its ability to engage and empower individuals to assist in transforming perspectives on pressing community concerns and inspire actions for positive public health change [6,10].

Many marginalized populations have limited ways to articulate their perspectives, and photovoice provides a creative approach to this effort. Since the 1870s, conventional media have often presented stereotypical images of the Appalachian region that construct a perception of the population as “other” [11]. Traditional mechanisms of expression may not be as viable or accessible, and the use of alternative, self-generated media is particularly salient in Appalachia. In addition, Appalachian scholars have noted that media act as “gatekeepers” that may suppress the concerns and actions of relatively powerless groups through degrading coverage or silence [12,13]. Encouraging rural youth, particularly from under-resourced communities like Appalachian Kentucky, to creatively express their perspectives on risks to environmental health has the potential for empowerment of youth and enlightenment for adults. Taking a photograph and writing about the captured image provides an individual’s perspective of their everyday lived experience. Photovoice, based on health promotion principles, can shape public policy around issues of environmental health [5]. According to Wang, photovoice has three main goals that enable participants to: “(1) record and represent their everyday realities; (2) promote critical dialogue and knowledge about personal and community strengths and concerns; and (3) reach policy makers” to achieve social change (p. 146, [5]). These three goals guided our research focus: to encourage youth living in an Appalachian community in which respiratory illness is common to explore what they perceive as contributing to respiratory burdens in their environments.

### 2.1. Participants: Eligibility, Recruitment, and Human Subject Protection

Eligibility criteria included young people living in Appalachian Kentucky, ages 12–18 years. Recruitment occurred primarily through referrals from CAB members, local nonprofit organizations, and teachers from the local high school. With human subject approval received by the University of Kentucky Institutional Review Board (15-0091-P6G), we administered consent and assent forms for the parents/guardians and youth, respectively. We attempted to prevent any potential risks by holding all sessions in a private room in a community location and by inviting parents or guardians to stay for all content if they wished.

### 2.2. Photovoice Protocol

Participants were invited to participate in three 2 h sessions, described below, as well as a culminating community exhibit. Two investigators with substantial CBPR experience led the sessions, with input from the local research coordinator. All study activities took place within an eight-week period and occurred in the evenings, with supper served. Parents were invited to listen to each session and join for supper, and many did join the gatherings. The community exhibit occurred two months after the sessions concluded.

In the first session, participants completed a pre-survey that measured their knowledge of respiratory illness causes and triggers and included open-ended questions asking about their reasons and expectations for participation. Knowledge was measured with multiple choice questions and included questions about the definition of respiratory illness; identification of asthma symptoms; triggers for wheezing and coughing; how local respiratory illness rates compare with other areas of the United States and reasons for the differences; and whether there are ways to reduce indoor lung disease triggers. The questions were repeated on the post-survey instrument. Youth were provided basic information about environmental factors that may influence or exacerbate respiratory disease that may be found inside and outside their homes, including the content of the knowledge questions in the pre-survey. Each participant was given a digital camera and a journal. They received instructions as to what they should photograph inside and outside their homes using the following question as their guide: “What factors inside and outside your home do you believe cause or trigger lung disease?” Participants were asked to take five photographs inside and outside their homes. In addition, they were asked to write short narratives in the journals to describe their photographs. Specifically, participants were instructed to write a few sentences explaining why they perceived the factor(s) captured in each photograph could cause lung disease. They were asked to elaborate as to (1) why they took the photograph of the factor(s) and (2) how the factor(s) might represent a cause for lung disease, and what might be a solution for change. An example photo and adjoining narrative was provided to participants.

**Y**outh were also provided basic instructions on how to capture an impactful photograph by a local resident who is a nationally recognized photographer and founder of the *Humans of Appalachia* project, Malcolm J. Wilson. His expertise and familiarity with the community provided the teens with a further sense of importance in what they were being asked to undertake. The training he provided included (1) capturing and framing a story; (2) lighting; (3) how to hold and use a camera; (4) the ethics of capturing an image; and (5) critiquing the photographic story. Participants were asked to protect their families’ privacy by not including their faces in photos. All sessions were audio recorded and transcribed.

### 2.3. Debriefing, Discussion, and Narratives

In session two, participants uploaded their photographs to a laptop computer, so they could be projected, shared with the group, and discussed. After all photos were uploaded, participants and investigators discussed the ease and challenges of photovoice. While Wang [5] integrated the SHOWeD method to facilitate discussion among participants, input from our local research coordinator and others indicated a need for a simplified guide [14]. Thus, while the questions that guided the discussion were aimed at achieving similar outcomes as the SHOWeD facilitation guide, including “identify the problem or the asset, critically discuss the roots of the situation, and develop strategies for improving the situation”, (p. 190, [5]), we modified the wording to be more age appropriate.

Participants volunteered to review their photos and narratives with the group. A local community member, also a photographer, guided the critique of the imagery, and investigators guided the discussion, making sure participants addressed questions on environmental perceptions of cause and solutions for change. The photovoice discussion included the following specific questions: (1) What did they see and write about; (2) How did the images and words represent their daily lived experiences; (3) How did the images represent environmental health challenges; and (4) What could be done to improve environmental health? After the group discussion and critique, the teens were given the opportunity to reshoot or add to their photographs and narratives.

At the third and final session, participants were asked to choose one or two favorite photographs and read their narratives to the group. Participants and researchers discussed common ideas that emerged among the chosen photovoice projects. In addition, we solicited discussion about what next steps should be taken for the project. Some participants thought the project should continue encouraging additional engagement from more young people, noting how they “felt their ideas mattered.” Each participant completed a post-survey that measured knowledge regarding environmental determinants of respiratory illness and open-ended questions regarding their photovoice experience. During the post survey, the participants were also asked to write how they can and will use the information they have gained to be catalysts for environmental health change in their community. This question in the post-survey was directly linked to the discussion during session three. Youth participants who completed the study were allowed to keep their cameras, camera cases, and journals.

### 2.4. Analysis

Analysis occurred in three stages. First, in the final photovoice session, participants selected and discussed their photographs and narratives to highlight their perspectives on environmental determinants of respiratory illness. This “selecting” stage is part of the participatory approach [6,10]. The second stage of analysis also occurred during the final session discussion. Participants were encouraged to identify and discuss similarities and differences among their projects, thus noting potential themes for analysis. The final stage occurred after completion of the sessions and the community exhibit. Open ended comments in the pre- and post-surveys, session discussion audio transcripts, and written notes were analyzed using content analysis to identify themes and codes recognized during stages one and two. One researcher collated all data and provided the other researchers with the initial code book of possible themes and collated data. Two other researchers analyzed the data, and inter-coder reliability was determined by comparing segments of texts related to the themes identified by researchers [15]. Discrepancies were discussed and resolved within the team, and consensus was achieved. The data used for analysis included (1) the hand-written notes that supplemented all audio recordings provided by the research staff, which included non-verbal cues among participants; (2) all audio transcripts, coded by three independent reviewers to ensure congruence in coding and frequency counts for utterances in sessions and narratives were cross-checked for consistency once each reviewer had completed their full analysis; (3) the final photovoice projects; and (4) the comments from the community exhibit. All data components were integral for the analysis and incorporated into the final results.

### 2.5. Dissemination

The Mountain Air Photovoice Project Community Exhibit occurred at a gallery in an arts center in Whitesburg, Kentucky. The purpose of the exhibit was to provide a community platform to bring attention to environmental health; encourage teens to get involved in environmental health, including presenting their ideas for solutions; and begin to craft next steps for community action. The invitation list was generated by investigators with input from the CAB and included the participants and their families and friends, members of the CAB, university partners, and other community stakeholders. Although elected officials were invited to the exhibit, they were not able to attend. However, the exhibit continues to be shared at community events in order to continue enhancing awareness of environmental determinants of respiratory illness. A comment card was available to attendees at the community exhibit asking for specific feedback regarding the photovoice projects and the exhibit experience.

## 3. Results

Seventeen teens participated in the first session, nine in the second session, and five in the final session. Reasons for participants not attending second and third sessions included employment obligations, school-related activities and inability to obtain transportation to the destination. Among the youth participants, all were white (consistent with the demographics of the county [16]), 71% were female, 11% reported an asthma diagnosis, and 71% received reduced or free lunch at school. The mean age was 15.7 years, and the mean grade level was 10.3 (range 8–12th grade). Ten children completed the pre- and post-surveys and submitted photos with adjoining narratives.

In the pre-survey written questionnaire, when asked about why participants might be interested in participating in the study, the most common theme described was “wanting to have a voice” and be involved in improving the environmental health of their community. Recognizing that the air quality is “not good”, participants wanted to learn why and “learn how to live a better life” for themselves and their families as they realized that commercially purchased products can be triggers for respiratory illness. One participant stated: “I want to learn about my own family’s respiratory illness”, and another participant said: “young people can make a change in the environment.” “I know that the air quality in my area is poor, and I would like to learn more about it. Also, a lot of people in my family have respiratory issues. I feel that photography is a good way to raise awareness.” “I love to take pictures and this way, I can do what I like to do, but at the same time I can show people what can cause their health harm.” Echoing many of the teens, they felt empowered to be invited to participate and encouraged that they could be perceived as agents of change.

Nineteen photovoice entries were completed by ten participants and were showcased at two exhibits—one in the originating community and one at an academic conference. Prior to both of these events, the CAB members were presented with the findings of the study and asked to offer feedback to investigators. Three overarching themes emerged from the photos and narratives. First, Appalachian youth captured the compromises community members make regarding respiratory health in order to secure a livelihood. Second, photos and narratives revealed the tension between cultural legacies and respiratory health. Third, images and captions highlighted respiratory consequences of geographic influences.

### 3.1. Compromising Health for Livelihood

Photos and narratives produced by Appalachian youth revealed their awareness of the simultaneous necessity and harm community members negotiate in order to secure a livelihood. Participants captured photos of exposures including coal dust or aerosol sprays that may include irritants. These exposures are commensurate with employment as miners, painters or custodians. Participants recognized the importance of employment and acknowledged the potential harm of these exposures in work environments. Photographing the workshop where her father had labored for years in order to provide income for her family, one participant remarked:

“My dad has a fabrication shop in our building and inside he uses spray paints. [The shop] isn’t ventilated.” (Figure 1).

Aware of the unique and omnipresent influence of the coal industry on the economics of Appalachian communities, another participant photographed the railroad track, located behind her house that used to transport coal regularly. Also acknowledging that exposures affect the health of workers and communities at large, she stated:

“These railroads are used to transport coal and in turn, coal dust moves from place to place. [Coal dust] lies in-between the tracks, as it does in the photo, and it travels through the air. Our area has been coal country for years, exposing us to things that people in most parts of the country are not exposed to. This photo represents those affected by lung disease that have worked in the coal industry and those in communities who may be exposed to the environmental effects.” (Figure 2).

In addition to documenting occupational hazards compromising respiratory health, participants noted other vital resources that threaten health. Though essential to survival in the winter, propane fireplaces are detrimental to household members’ lungs. Recognizing this tension, one participant noted about the fireplace in their home:

“We use propane to heat our house in the winter. When the fireplace is on, you can smell the propane and it is definitely not good to breathe. Propane takes the place of oxygen in the lungs. We use these [two silver propane tanks] in the winter…”.

### 3.2. Cultural Legacy

Through their photos and narratives, Appalachian youth documented the tension they observed between cultural legacies and respiratory health. Smoking is highly prevalent among Appalachian residents and deeply entrenched in local culture, and participants provided many examples of how common the practice is inside and outside their homes. One participant photographed a tin can for collecting cigarette butts that sits on the porch at her family’s home. She remarked:

“They use it as an ashtray…and I chose this picture because everybody in my family smokes, and they either have cancer or COPD.” (Figure 3).

Another participant wrote of her photo that depicts cigarettes, noting:

“A lot of people around here live in poverty, and they don’t have enough money to buy all the things that they should be buying…and they have addictions and they can’t help it and sometimes, cigarettes are more important to them than the other things.” (Figure 4).

Another cultural mainstay, which often co-occurs with smoking, is the common practice of family members frequently sitting on the front or back porch to socialize. One participant photographed and wrote about a citronella candle. She remarked:

“My family keeps these [citronella candles] around to ward off bugs. However, this is a pesticide that can cause harm to humans. Releasing chemicals in the air is extremely dangerous.” (Figure 5).

Finally, one participant noted that if she wanted to spend time with her family, she must endure exposure to smoke. Referring to her photo of a family member smoking, the participant remarked:

“This picture is of a person on my back porch. They are smoking, and I wanted this picture to bring some humanity into the smoking issue…it reminds us that people struggle with this issue almost everywhere in Eastern Kentucky.” (Figure 6).

### 3.3. Geographic Forces

In resource-limited communities, geographic and climatic realities can prove formidable challenges. Youth captured the influence geography can have on respiratory health in their region. Geographic forces included triggers that participants characterized as the result of the outside environment coming inside through ventilation such as a fireplace, air vent, mildew and mold, and dust (Figure 7).

Participants’ photographs and captions revealed their awareness of ubiquitous mold, mildew, and dust in their homes that come from inside and outside pollutants. Concerned about the impact of hidden but constant dust exposure on his and his family’s health, one participant wrote:

“I lifted up the vent cover in my house to reveal great amounts of dust and other garbage settled in the bottom and hanging from the cover. When the air is on, this material and particles of dust are circulating around our home and finding way[s] into our airways.” (Figure 8).

Through their narratives, many participants demonstrated how their personal exposure to respiratory hazards is echoed by respiratory risk factors permeating their whole community. For example, one participant discussed the boots they found in a storage cabinet:

“[These boots] were found in an old closet of my family’s. They are covered in mold and this mold is also found all throughout my house…a clue to the weather. These boots show the controlling nature of mold in my community.” (Figure 9).

### 3.4. Community Response

Approximately 30 individuals attended the community exhibit. Attendees included the participants and their families, MAP CAB members and other community members. The community exhibit provided a place and space for further distribution of the data. Comment cards were available on a table near the exhibit for those who wished to provide feedback. Six comment cards were completed. The comments supported the inclination to include teens in the photovoice project and that qualitative research, specifically visual imagery, can be powerful. Exhibit attendees characterized their perceptions of the youth participants’ efforts:

“Youth are more aware and intelligent than imagined.”

“[Looking] through the lens of a camera taught our youth to be more observant.”

When asked about which messages surprised attendees, responses included:

“[The] number of photos dedicated to tobacco.”

“Drawing attention through photos makes it much worse.”

Exhibit attendee responses to the photovoice project provided further insight into the theme of cultural legacy. First is the acknowledgment that youth have important contributions, and second, that viewing the tobacco photographs, taken by the teens, draws attention to the issue in a different way than an advertisement might or reading about the harms of smoking.

## 4. Discussion

Incorporating the perspectives of youth as characterized by the photographs and accompanying narratives proved valuable in this exploratory study. Youth participants exceeded the expectations of investigators to contribute agency and ideas to improving environmental health. At the conclusion of the study, several participants informed investigators of their desire to disseminate their findings: One said: “[you] need to share [the photovoice project] with people in the community.” Another said: “I feel proud to have been a part of something so active. I hope it will result in change.” Photovoice participants presented their completed work to the CAB and at a research conference, stating that their experiences in MAP influenced their desire to pursue research as a career. Youth participants’ photographs contributed to identification of environmental factors that investigators had not measured in the original MAP study, including aerosol sprays, propane tanks and pollen. These findings may inform the next iteration of the MAP survey instrument. Furthermore, these findings reinforce the importance of including youth in CBPR in Appalachian communities. Not only did the findings demonstrate the capacity of youth to contribute to environmental health research, but their participation also resulted in increased community awareness of youth perspectives and appreciation for youth agency.

While community–academic partnerships to address environmental health challenges are not new, to our knowledge, this was the first youth-led CBPR study to address environmental health in Appalachia. Youth are traditionally excluded or overlooked from social action or research engagement, and a limited number of published studies have been youth-led [17,18]. We could identify only three published participatory research studies that involved youth in Appalachia; these studies focused on obesity/diabetes [7] and sugar-sweetened beverage consumption [8,9]. However, this project may be the first to equip youth in Appalachia with sufficient agency to guide the intervention development and formative assessment for an environmental health focus.

Several strengths contributed to the success of this exploratory study. First, based on the comments provided by participating youth, investigators discovered that the participants had previously been offered limited opportunity to contribute to community health initiatives. Youth expressed feelings of empowerment through their participation in this project, and nearly 59% of the youth who began the study completed it. Inviting parents to stay for the sessions, and encouraging their involvement strengthened the engagement of those youth whose parents/guardians did stay. Many parents/guardians asked questions during sessions and attended the community exhibit. Finally, having a local research coordinator who knew many of the families was critical for the recruitment and retention of participants, as the study site was located nearly three hours away from the university.

This research also has notable limitations. Elements of this study are region-specific, particularly participant concerns related to coal mining, and may not be generalizable outside Appalachia. However, this limitation may be balanced by the commonalities between Appalachia and other disadvantaged regions. Many of the environmental health challenges that youth recognized, such as the competing demands of health and occupation, are relevant in many communities that experience environmental injustice. Second, this study did not explicitly seek policy change, which differs from many photovoice projects. Instead, we found that the participants were most interested in using the project to advocate for positive changes within their own families first. Participants’ photos allowed them to question adult choices and consider healthier alternatives for themselves. In a region with a high prevalence of tobacco use and miner’s pneumoconiosis [19,20,21], youth advocacy may offer a powerful strategy for improving respiratory health through personal and environmental change.

Sustaining youth participation presented some challenges. The study occurred during the school year, so meetings were held in the evening on a weeknight in a community location. Regular participation was difficult for some participants due to conflicts with afterschool activities, work obligations, or a lack of transportation. Communication was also challenging, as participants initially requested the use of a Facebook group page to communicate, but later failed to respond to communication from study personnel. Text messages and phone calls were also not productive to yield responses. Future research may yield enhanced communication and participation if the study is embedded within a class or a student organization so that sessions can take place during the school day.

This study elicited Appalachian youth-identified targets for improving environmental health. Initial dissemination to the community demonstrated the power of youth voices, and findings from this study will inform the development of a home-based intervention to improve respiratory illness symptoms.

## 5. Conclusions

Photovoice is an important CBPR approach to not only collect qualitative environmental health data but also a feasible means by which to actively engage youth. In this Appalachian community, participating youth demonstrated capacity to engage in research and an eagerness to contribute their perspectives to environmental health matters. As we continue to examine contributing factors to the extensive health disparities in this region, this study demonstrated the value of including the voices of youth in future environmental health research.

## Figures and Tables

**Figure 1 ijerph-16-03829-f001:**
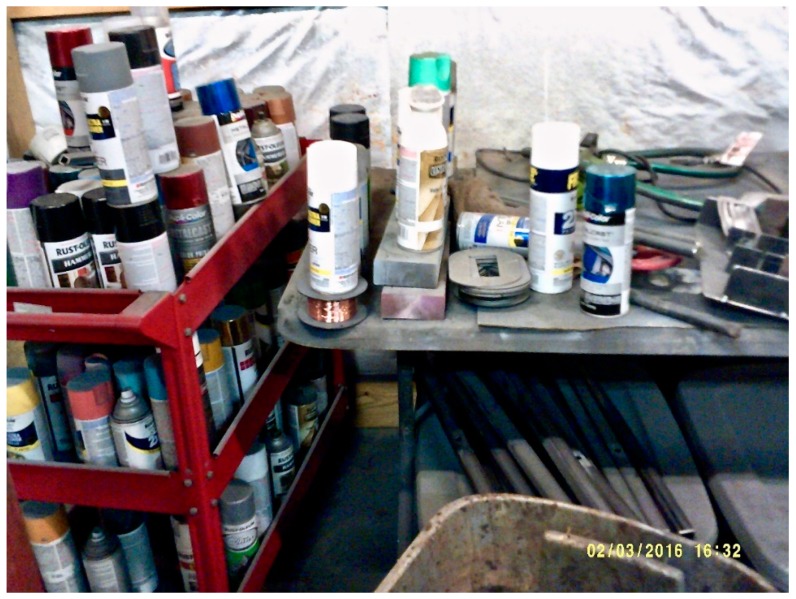
Spray paints in fabrication workshop.

**Figure 2 ijerph-16-03829-f002:**
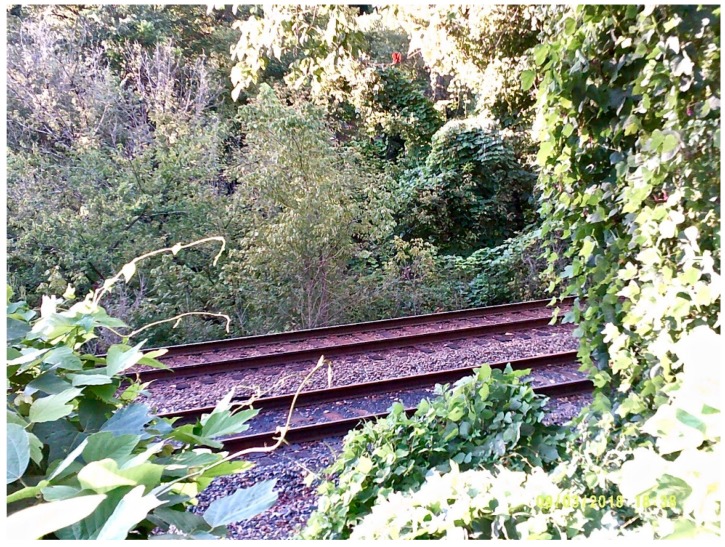
Railroad used for coal transport.

**Figure 3 ijerph-16-03829-f003:**
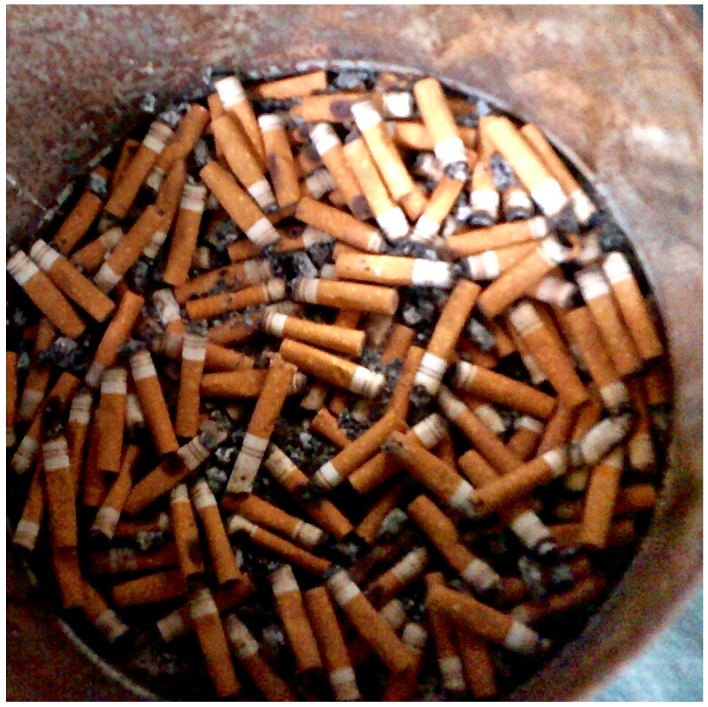
Ashtray.

**Figure 4 ijerph-16-03829-f004:**
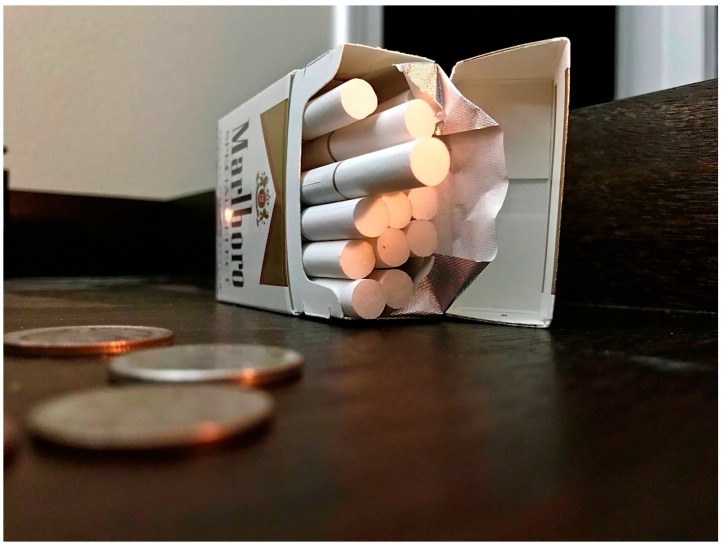
Cigarettes and their cost.

**Figure 5 ijerph-16-03829-f005:**
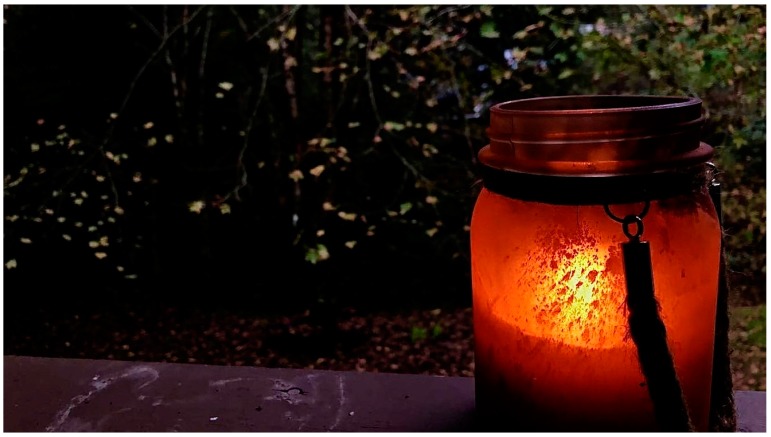
Porch candle.

**Figure 6 ijerph-16-03829-f006:**
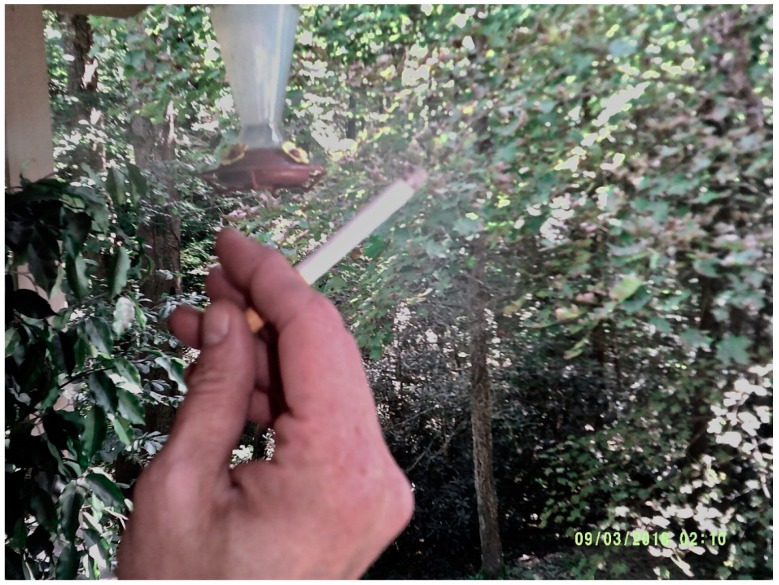
Cigarette smoking on the back porch.

**Figure 7 ijerph-16-03829-f007:**
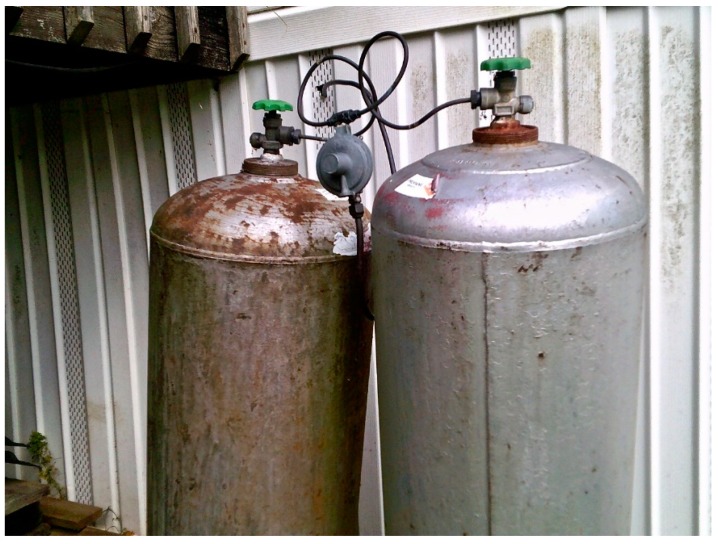
Propane tanks.

**Figure 8 ijerph-16-03829-f008:**
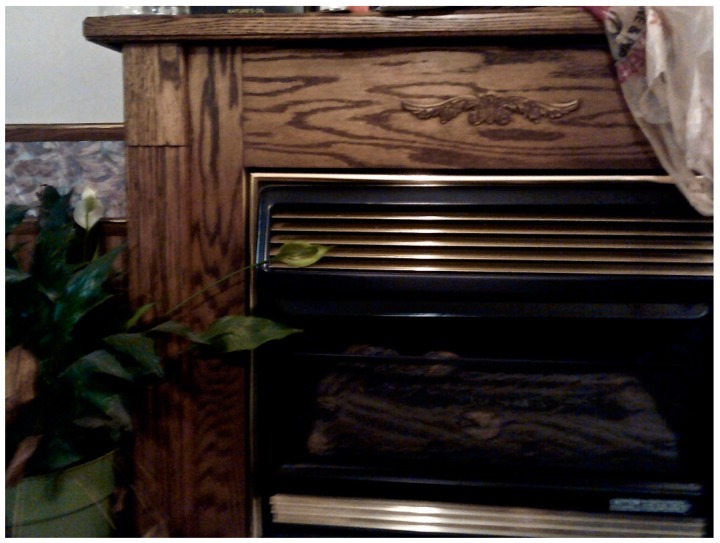
Dust around vents.

**Figure 9 ijerph-16-03829-f009:**
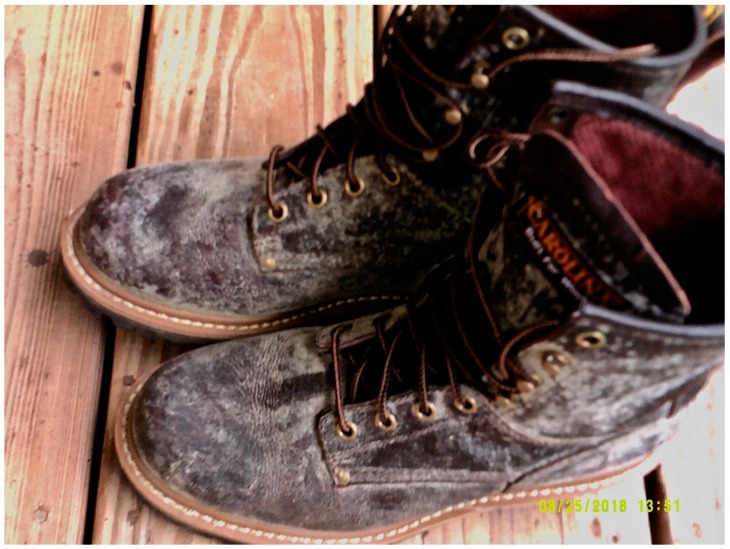
Mold on boots.

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
