# Peer review of "“Youth Are More Aware and Intelligent than Imagined”: The Mountain Air Youth Photovoice Project"

_ijerph, 2019, doi:10.3390/ijerph16203829_

Round 1
Reviewer 1 Report
This is a interesting project and has important implications for empowering youth. I have a few comments mainly centering providing greater detail on the analysis.
Could you provide a more background on the MAP project. What are its goals and how does the present study fit in with those goals. It would be helpful to put into context. What is the goal of the exploratory study, what are the next steps? This was missing in intro and in discussion. Cite previous studies done with Appalachian youth when mentioned in introduction. Under 2.2, could you provide more info on research team, who led the sessions? How many people? The Analysis of photo voice projects is often complex and it would be helpful to see more details on how the analysis was conducted. Under 2.4, the paper states that participants selected and discussed their photographs. FIRST STAGE OF ANALYSIS: How were photos selected and how was discussion guided? What was the process the research team used to guide students in picking out their pictures and in their discussion. What questions guided the discussion? SECOND STAGE: What questions were used to help participants identify similarities and differences across projects? Did they come up with themes from the analysis? How were researchers involved in deciding on the final themes. How much were researchers involved in this discussion and these decisions about final themes from the second stage of analysis. It would be helpful for others who wish to learn from your process to hear more about how you guided the analysis. THIRD STAGE: The final stage was an analysis of open-ended comments, discussion transcripts, and narratives (when were these narratives collected?). It is stated that a qualitative method was used. What qualitative method was used? open-coding? thematic analysis? What method did you follow? Cite it. All transcripts were coded by independent coders, so what process was used to ensure congruence in coding? Be specific. What exactly did they code? Frequency counts? Frequency counts of what? Results: Three themes came out of the analysis--the analysis needs to be clear about how the final three themes emerged. How was validity and reliability ensured regarding the emergence of the three themes. Three stages of analysis are mentioned--how were they all integrated? and then how did the lead to the results discussed?Author Response
This is a interesting project and has important implications for empowering youth. I have a few comments mainly centering providing greater detail on the analysis.
Thank you for your thoughtful suggestions for improving the paper. We believe the suggested changes improve the manuscript substantially.
Could you provide a more background on the MAP project. What are its goals and how does the present study fit in with those goals. It would be helpful to put into context.
We have added a paragraph in the introduction to provide additional information about MAP and how this study fits with the broader MAP initiative.
What is the goal of the exploratory study, what are the next steps? This was missing in intro and in discussion.
In the introduction section (second paragraph), we attempted to clarify that the goal of this exploratory study was to obtain grounded perspectives on what youth considered contributors to respiratory disease and to gauge their awareness of perspectives on well-known and emerging contributors using photovoice. In the discussion section (at the end), we specified that findings from this study will inform the development of a home-based intervention to improve respiratory illness symptoms.
Cite previous studies done with Appalachian youth when mentioned in introduction.
We have added these references to the introduction.
Under 2.2, could you provide more info on research team, who led the sessions? How many people?
We added a sentence in this section to specify that two investigators led the sessions, with input as well from the research coordinator.
The Analysis of photo voice projects is often complex and it would be helpful to see more details on how the analysis was conducted. Under 2.4, the paper states that participants selected and discussed their photographs. FIRST STAGE OF ANALYSIS: How were photos selected and how was discussion guided? What was the process the research team used to guide students in picking out their pictures and in their discussion. What questions guided the discussion? SECOND STAGE: What questions were used to help participants identify similarities and differences across projects? Did they come up with themes from the analysis? How were researchers involved in deciding on the final themes. How much were researchers involved in this discussion and these decisions about final themes from the second stage of analysis. It would be helpful for others who wish to learn from your process to hear more about how you guided the analysis. THIRD STAGE: The final stage was an analysis of open-ended comments, discussion transcripts, and narratives (when were these narratives collected?). It is stated that a qualitative method was used. What qualitative method was used? open-coding? thematic analysis? What method did you follow? Cite it. All transcripts were coded by independent coders, so what process was used to ensure congruence in coding? Be specific. What exactly did they code? Frequency counts? Frequency counts of what? Results: Three themes came out of the analysis--the analysis needs to be clear about how the final three themes emerged. How was validity and reliability ensured regarding the emergence of the three themes. Three stages of analysis are mentioned--how were they all integrated? and then how did the lead to the results discussed?
In sections 2.2., 2.3., and 2.4., we have substantially revised the methods to provide greater details and clarity as to the protocol that we used in implementing and analyzing photovoice.
Reviewer 2 Report
Kudos to the authors for completing this important project and drafting a well-written paper. I have a few important suggestions that might improve your article.
Title…I think that the title of the article is a little misleading in regards to the word “change.” Although a core goal of photovoice is to advocate for policy change, that did not happen at the end of this project. Please edit that part of your title so that it is a more accurate reflection of the project
Introduction…the “p” and “v” should not be capitalized in photovoice
Introduction…lines 49-50…do not bold the sentence
Introduction…lines 48-49…please cite which studies you’re referring to “Whereas PhotoVoice has been used in previous studies with Appalachian youth…” I know that you cite these in the discussion section, but it would be helpful to see these in the introduction.
Introduction…please add another paragraph that describes what photovoice is and how it works
Materials and methods…lines 103-105… do not bold the sentence or the “y” of the next sentence
Materials and methods…lines 118-122… I’m assuming that the discussion questions were based from the “SHOWeD” mnemonic. If so, I’d cite one of Wang’s early articles that describe SHOWeD and state that your questions were based on it
Materials and methods…a typical component of photovoice is to have participants write a caption for each photo, which is then displayed with the photos. Did the participants write any captions for this project? Is this what you’re referring to with the word “narrative”? If so, please describe how you guided the participants through the caption writing process. That tends to be the most frustrating and difficult part of my own photovoice projects, so I'm curious what your experience was like.
Materials and methods…line 153…how did you determine which stakeholders to invite? Were any of those stakeholders community policymakers? The third goal of photovoice (and in my opinion it’s the most important) is to reach policymakers so that participants can advocate for policy change. Regardless if change happens (policy change is very difficult), it's important to at least reach policymakers and attempt to advocate for change. Did that take place? If not, why not?
Discussion…please add a paragraph or two regarding any actual change that happened as a result of the project. It’s great that the participants identified a few issues that the researchers didn’t think of before (e.g., spray paint, propane, pollen), but what else happened as a result of the project? The discussion section needs to be strengthened in that regard. If that's the only change that happened, then please edit that paragraph so that the importance of spray point, propane, etc., stands out more to those reading your article.
Discussion…please include a limitations section to your article (e.g., generalizability, lack of advocacy for policy change). You might want to move the last paragraph of your discussion within the limitations section.
Author Response
Kudos to the authors for completing this important project and drafting a well-written paper. I have a few important suggestions that might improve your article.
Thank you for comments, and we appreciate your thoughtful suggestions that have improved the paper.
Title…I think that the title of the article is a little misleading in regards to the word “change.” Although a core goal of photovoice is to advocate for policy change, that did not happen at the end of this project. Please edit that part of your title so that it is a more accurate reflection of the project
We agree that the quote that was selected for inclusion in the title was not adequately reflective of the manuscript content and have modified the title. We chose to include a different quote in the title, now reading, “Youth are more aware and intelligent than imagined”: the Mountain Air Youth Photovoice Project.
Introduction…the “p” and “v” should not be capitalized in photovoice
We have changed all mentions of “PhotoVoice” to photovoice.
Introduction…lines 49-50…do not bold the sentence
We have removed the bold font for the sentence.
Introduction…lines 48-49…please cite which studies you’re referring to “Whereas PhotoVoice has been used in previous studies with Appalachian youth…” I know that you cite these in the discussion section, but it would be helpful to see these in the introduction.
We have added the references in the introduction.
Introduction…please add another paragraph that describes what photovoice is and how it works
We added a paragraph in the introduction that provides additional information about what photovoice is and how it works.
Materials and methods…lines 103-105… do not bold the sentence or the “y” of the next sentence
We have removed the bold font for the sentence.
Materials and methods…lines 118-122… I’m assuming that the discussion questions were based from the “SHOWeD” mnemonic. If so, I’d cite one of Wang’s early articles that describe SHOWeD and state that your questions were based on it
Yes, the discussion questions were based on the “SHOWeD” mnemonic. We added additional description in the methods to reflect this and added a citation from Wang.
Materials and methods…a typical component of photovoice is to have participants write a caption for each photo, which is then displayed with the photos. Did the participants write any captions for this project? Is this what you’re referring to with the word “narrative”? If so, please describe how you guided the participants through the caption writing process. That tends to be the most frustrating and difficult part of my own photovoice projects, so I'm curious what your experience was like.
Thank you for pointing out this omission. We added description of the narratives that participants provided in their journals as well as the instructions with which participants were provided as to how to construct their narratives.
Materials and methods…line 153…how did you determine which stakeholders to invite? Were any of those stakeholders community policymakers? The third goal of photovoice (and in my opinion it’s the most important) is to reach policymakers so that participants can advocate for policy change. Regardless if change happens (policy change is very difficult), it's important to at least reach policymakers and attempt to advocate for change. Did that take place? If not, why not?
We added a description as to how the exhibit invitation list was generated. While elected officials in this small community were invited to attend the community exhibit, they were unable to attend, perhaps because it was held on a Sunday afternoon. However, we are confident that some elected officials have viewed the photos that continue to be shared in the community. We added this information in section 2.5 under dissemination.
We agree with the reviewer’s views regarding the importance of policy change as action resulting from CBPR. In this study our goal was less oriented to policy change but rather to enhance awareness of factors, some of which are modifiable, that contribute to respiratory illness in and around homes. We also added a reference to this in the limitations paragraph in the discussion section.
Discussion…please add a paragraph or two regarding any actual change that happened as a result of the project. It’s great that the participants identified a few issues that the researchers didn’t think of before (e.g., spray paint, propane, pollen), but what else happened as a result of the project? The discussion section needs to be strengthened in that regard. If that's the only change that happened, then please edit that paragraph so that the importance of spray point, propane, etc., stands out more to those reading your article.
In the first paragraph of the discussion, we clarified the impact of the study on the community in terms of enhanced community awareness of youth perspectives and youth agency. Additionally, we discussed the type of change that participants sought from the study (i.e., within their own families). Finally, we added a paragraph at the end of the discussion to note the next steps for our research, based on this study’s findings.
Discussion…please include a limitations section to your article (e.g., generalizability, lack of advocacy for policy change). You might want to move the last paragraph of your discussion within the limitations section.
We added a paragraph in the discussion section to describe limitations of our study.